# The Crosstalk between N-Formyl Peptide Receptors and uPAR in Systemic Sclerosis: Molecular Mechanisms, Pathogenetic Role and Therapeutic Opportunities

**DOI:** 10.3390/ijms25063156

**Published:** 2024-03-09

**Authors:** Filomena Napolitano, Francesca Wanda Rossi, Amato de Paulis, Antonio Lavecchia, Nunzia Montuori

**Affiliations:** 1Department of Translational Medical Sciences, University of Naples Federico II, 80138 Naples, Italy; filomena.napolitano@unina.it (F.N.); francescawrossi@gmail.com (F.W.R.); depaulis@unina.it (A.d.P.); 2Center for Basic and Clinical Immunology Research (CISI), WAO Center of Excellence, University of Naples Federico II, 80131 Naples, Italy; 3“Drug Discovery” Laboratory, Department of Pharmacy, University of Naples Federico II, 80131 Naples, Italy; antonio.lavecchia@unina.it

**Keywords:** systemic sclerosis, urokinase receptor, N-formyl peptide receptors, fibroblast proliferation, small molecules

## Abstract

Systemic Sclerosis (SSc) is a heterogeneous autoimmune disease characterized by widespread vasculopathy, the presence of autoantibodies and the progressive fibrosis of skin and visceral organs. There are still many questions about its pathogenesis, particularly related to the complex regulation of the fibrotic process, and to the factors that trigger its onset. Our recent studies supported a key role of N-formyl peptide receptors (FPRs) and their crosstalk with uPAR in the fibrotic phase of the disease. Here, we found that dermal fibroblasts acquire a proliferative phenotype after the activation of FPRs and their interaction with uPAR, leading to both Rac1 and ERK activation, c-Myc phosphorylation and Cyclin D1 upregulation which drive cell cycle progression. The comparison between normal and SSc fibroblasts reveals that SSc fibroblasts exhibit a higher proliferative rate than healthy control, suggesting that an altered fibroblast proliferation could contribute to the initiation and progression of the fibrotic process. Finally, a synthetic compound targeting the FPRs/uPAR interaction significantly inhibits SSc fibroblast proliferation, paving the way for the development of new targeted therapies in fibrotic diseases.

## 1. Introduction

Systemic Sclerosis (SSc) is a complex chronic autoimmune disease that mainly affects connective tissue, microvessels and small arteries, and is characterized by fibrosis and vascular obliteration in the skin and internal organs, particularly the lungs, heart and digestive tract [1,2]. The pathogenesis of SSc is extremely complex, and, despite numerous studies, the mechanisms involved in its development and maintenance are still not completely clarified.

During the past decade, most of the studies have been focused on the role of myofibroblasts in the pathogenesis of SSc [3]. Myofibroblasts are fibroblasts with contractile properties that release a large amount of profibrotic extracellular matrix (ECM) molecules, mainly collagen type I [4,5,6]. In 1972, it had already reported that fibroblasts obtained from SSc skin have a profibrotic phenotype and produce more collagens than control fibroblasts [7]. Subsequently, it was demonstrated by immunohistochemistry analysis that fibroblasts isolated from lesional areas of skin, the esophagus, and lungs from SSc patients express high levels of alpha smooth muscle actin (α-SMA), a typical marker of the myofibroblast phenotype [8].

In SSc, several mechanisms protect myofibroblasts from apoptosis, ensuring their continued presence after their formation [9]. First, it has been observed that, in the quiescent state, SSc myofibroblasts express less proapoptotic BAX compared to myofibroblasts from healthy controls [10]. Secondly, in dermal fibroblasts isolated from patients with SSc, the hyperactivation of PI3 kinase/AKT signaling facilitates myofibroblast survival through the inhibition of BAX activity [11]. Recently, it has been reported that epigenetic modifications can also regulate the evasion of apoptosis in myofibroblasts and specific micro RNAs (miRNAs) have been reported to be involved in SSc fibrosis [12,13].

The innate immune system plays an important role in SSc myofibroblast survival, formation, and function. Cells that stimulate myofibroblast functions, through the release of profibrotic cytokines, IL-4, IL-13 and TGF-β, include mast cells, monocytes/macrophages and T helper 2 lymphocytes [9,14,15]. Recent advances have demonstrated the involvement of B cells in SSc-related fibrosis, as well as in the immune abnormalities. B cell depletion therapy with rituximab has improved the extent of skin sclerosis evaluated using the modified Rodnan skin thickness score and a skin biopsy at 12 weeks has shown a significant reduction in myofibroblasts. Importantly, high-affinity topo I-reactive B cells, via the differentiation of T cells into Th17 cells, can stimulate fibroblasts to release collagen and pro-inflammatory cytokines [16].

Over the last decade, we have focused on the role of a group of innate immune receptors, N-formyl peptide receptors (FPRs), in the pathogenesis of SSc. FPRs are G protein-coupled chemoattractant receptors that play important roles in host defense and inflammation [17]. They are so called because their cognate agonists are peptides containing formylated methionine (fMet), such as those derived from bacterial proteins. Three FPRs have been identified in humans, namely FPR1, FPR2 and FPR3 [18]. Among all the ligands for the FPRs, fMet-Leu-Phe fMLF (fMLF), a potent leukocyte chemoattractant, is most often studied. FPR1 and FPR2 are, respectively, a high affinity receptor and low affinity receptor for fMLF, while FPR3 does not bind fMLF. The FPR expression on epithelia seems to be required for wound repair and the restitution of barrier integrity. In this context, it has been demonstrated that FPR activation promotes cell migration, proliferation, and neoangiogenesis in intestinal, lung, retinal pigment, and nasal epithelial cells [19,20].

Many functions of FPRs occur through their interaction with the urokinase-type plasminogen activator receptor (uPAR) [21,22,23]. uPAR, a glycosyl-phosphatidyl-inositol-(GPI)-anchored protein formed by three domains (DI-DII-DIII), serves to bind the urokinase plasminogen activator (uPA) and localize the reactions of the plasminogen activation system on the cell membrane [24,25,26,27]. After removal of the GPI anchor by proteases or phospholipases, uPAR sheds from the cell membrane and exists as a soluble form (suPAR) that is detectable in various body fluids [28]. uPA binding promotes the clustering of uPAR in the cell membrane and increases its ability to bind vitronectin (VN), which is associated with the provisional matrix in cancer and inflammation. In addition, uPAR interacts with different transmembrane receptors, including integrins, the epidermal growth factor receptor (EGFR), and the platelet-derived growth factor receptor (PDGFR). Both membrane and soluble uPAR can expose a specific region, corresponding to amino acids 88–92 (SRSRY), able to interact with FPRs and mediating uPA or fMLF-dependent cell migration. The exposure of the SRSRY sequence is determined by a conformational change in the receptor, upon binding to uPA, or cleavage of D1 by metalloproteases or uPA itself, thus leading to the formation of DII-DIII-uPAR_88–92_ [29].

Unaccomplished data on the role of uPAR in the pathogenesis of SSc have been reported in the literature. Manetti et al. have shown that uPAR-deficient (uPAR^−/−^) mice represent a model of experimental SSc as the inactivation of the uPAR gene induces dermal and pulmonary fibrosis and peripheral microvasculopathy. Moreover, the skin of SSc patients exhibits a significantly decreased expression of native full-length uPAR [30]. In these studies, however, only the full-length uPAR, and not the cleaved form, was analyzed. Recently, we have demonstrated, by immunohistochemical analysis, that SSc skin fibroblasts overexpress all three isoforms of FPRs and DII-DIII-uPAR_88–92_, while full-length uPAR expression is downregulated. Our data assigned a crucial pathogenetic role to DII-DIII-uPAR_88–92_ in SSc as the overexpression of cleaved uPAR and its relationships with FPRs correspond to the dysfunctional activities of fibroblasts in the development of fibrosis. Indeed, we have demonstrated that FPR activation can induce a myofibroblast phenotype in normal dermal fibroblasts through α-SMA induction. Both in normal and SSc fibroblasts, FPRs/uPAR crosstalk enhances the rate of wound healing, ECM deposition and the generation of reactive oxygen species (ROS), that play a key role in the alteration of the redox state observed in SSc. Even more interestingly, C37, a new small molecule able to inhibit uPAR binding to FPRs, can inhibit ROS production in SSc fibroblasts [22,31].

Starting from this experimental evidence, we now analyze the effects of FPR activation and crosstalk with uPAR on dermal fibroblast proliferation to elucidate their molecular mechanisms and signaling pathways. The role of FPRs/uPAR crosstalk, already proven to be important in fibroblast-to-myofibroblast transition and ROS production, will also be studied in the proliferation of fibroblasts isolated from an SSc skin biopsy to propose new therapeutic strategies.

## 2. Results

### 2.1. FPR Activation Promotes Proliferation in a Normal Human Dermal Fibroblast Cell Line

Fibrosis represents the hallmark of Systemic Sclerosis (SSc), the end stage triggered by different pathological events [32].

There is evidence linking the urokinase receptor (uPAR) with N-formyl peptide receptors (FPRs), both highly expressed in SSc patients, and suggesting that it may have a central role in fibrosis and in fibroblast-to-myofibroblast transition [22,31].

Since the proliferation of skin fibroblasts is a key factor in SSc dermal fibrosis, we investigated the role of uPAR and FPRs in skin fibroblast proliferation. To this aim, we evaluated cell proliferation at 0, 24, 48, 72 and 144 h after stimulation with specific FPR agonists. The proinflammatory chemoattractant fMLF was used at two concentrations 10^−4^ M and 10^−8^ M because FPR1 and FPR2 have high and low affinity, respectively, for fMLF [18]. The synthetic WKYMVm hexapeptide was used at a concentration of 10^−8^ M and the synthetic soluble uPAR_84–95_ peptide, containing the uPAR-derived ^88^SRSRY^92^ sequence and able to interact with FPRs on the cell surface and to activate their signals, was used at 10^−8^ M. In these preliminary studies, the BJ cell line has been used as a human dermal fibroblast model.

As shown in Figure 1A, all the agonists were significantly able to induce BJ proliferation, thus suggesting that both FPRs and FPRs/uPAR interaction are involved in the regulation of fibroblast proliferation. fMLF 10^−4^ M significantly enhanced fibroblast proliferation at 24, 48 and 144 h; fMLF 10^−8^ M significantly induced the fibroblast proliferation at 24, 48 and 72 h, exerting no effects at 144 h. Importantly, fMLF stimulation displayed a bell-shaped response curve typical for chemokines [33,34]. The WKYMVm synthetic peptide and the synthetic soluble uPAR_84–95_ exhibited a similar behavior. WKYMVm peptide significantly enhanced the proliferation at 24, 48, and 72 h, and uPAR_84–95_ peptide exerted its effects at 24 and 72 h.

Subsequently, we asked which isoforms of the FPR family are stimulated by the agonists used in previous experiments. To answer this question, we performed fibroblast proliferation assays in the presence of specific anti-FPR antibodies, following stimulation with agonists. Figure 1B shows that fMLF, at both a high (10^−4^ M) and low (10^−8^ M) concentration, was unable to elicit proliferation in the presence of antibodies directed against FPR1 and FPR2. As expected, in the presence of the anti-FPR3 antibody, cells still proliferated; indeed, FPR3 does not bind to fMLF. The WKYMVm synthetic peptide is mostly described as a FPR2 agonist [35]; interestingly, we observed that WKYMVm stimulation was also unable to elicit cell proliferation in the presence of anti-FPR3 antibodies, thus indicating the involvement of this receptor in FPR-mediated proliferative response. The synthetic soluble uPAR_84–95_ peptide behaved as a the WKYMvm peptide, confirming that uPAR_84–95_ induced basophil chemotaxis mainly by activating FPR3 and, to some extent, FPR2 [36].

From these experiences, we would highlight that: (i) FPR activation promotes dermal fibroblast proliferation; (ii) FPRs/uPAR crosstalk is involved in dermal fibroblast proliferation; (iii) uPAR, in the synthetic soluble form containing the uPAR-derived ^88^SRSRY^92^ sequence, is able to interact also with FPR3 isoforms on the cell surface.

### 2.2. Rac1 and ERK1/2 Activation by FPRs in Dermal Fibroblast Proliferation

uPAR-mediated cell migration, allowed by uPAR interactions with FPRs and β1 integrins, involves small GTPase Rac1 as signaling mediators [37]. In addition, FPR stimulation determines ROS production through the Rac1 and ERK1/2 signaling pathways [21]. uPAR-dependent signaling pathways may also lead to the activation of ERK1/2 MAPKs through the activation of PI3K [38]. Thus, to identify the signaling pathways involved in FPRs-induced proliferation, we focused on the small Rac1 GTPase and ERK1/2 pathways.

We analyzed dermal fibroblast proliferation after stimulation with optimal concentrations of fMLF (10^−4^ M), uPAR_84–95_ (10^−8^ M) and WKYMVm peptide (10^−8^ M). These analyses were performed in the presence of specific inhibitors: NSC23766 (25 µM), a Rac-specific GEF (guanine nucleotide exchange factor) Trio and Tiam1 inhibitor and selumetinib (2.5 µM), a specific MEK 1/2 inhibitor (Figure 2).

BJ cells were unable to proliferate in presence of selumetinib at all time points. Instead, NSC23766 solely inhibited fibroblast proliferation in response to uPAR_84–95_ (10^−8^ M) and WKYMVm peptide (10^−8^ M) at 72 h. These results demonstrated that FPR isoforms, after binding to ligands, trigger different signaling pathways, as already suggested in the literature [39,40]. The MAPK signaling pathway was directly activated by all three FPR isoforms whereas Rac1 GTPase seems to be downstream of FPR3 activation, since NSC23766 inhibited fibroblast proliferation at 72 h in response to uPAR_84–95_ and WKYMVm peptides, but not fMLF.

### 2.3. uPAR/FPRs Crosstalk-Dependent c-Myc Phosphorylation and Cyclin D1 Expression in Normal Human Dermal Fibroblasts

Upon activation, intracellular domains of FPRs mediate signaling to G-proteins, which trigger several signal transduction pathways, phosphorylation and the nuclear translocation of regulatory transcriptional factors, calcium release and the production of oxidant compounds [18]. Further studies are required to better define the intracellular signaling pathways triggered by FPR activation and crosstalk with uPAR.

Thus, we focused on the most significant intracellular pathways for cell cycle progression and cell proliferation, such as c-Myc phosphorylation and cyclin D1 induction, in response to FPRs/uPAR crosstalk activation in dermal fibroblasts, using uPAR_84–95_ peptide (10^−8^ M).

The c-Myc transcription factor is a potent regulator of cell growth, proliferation, apoptosis, differentiation, and metabolism [41]. c-Myc protein stability is regulated by phosphorylation at threonine 58 (Thr58) and serine 62 (Ser62) residues. Ser62 phosphorylation by cyclin-dependent kinases (CDKs) or extracellular signal-regulated kinase increases protein stability, while Thr58 phosphorylation by GSK3 promotes Ser62 dephosphorylation and targets c-Myc for degradation [42].

Cyclin D1 is a key intracellular mediator of extracellular signals, which regulates cell proliferation and is responsible for cell cycle progression in the transition from G0/G1 to S phase [43].

We observed that the treatment of fibroblast cells with uPAR_84–95_ peptide induced a time-dependent phosphorylation of c-Myc at Ser62 residue and upregulation of Cyclin D1, as compared to cells at T0 (Figure 3).

These results show that FPRs, beyond their role in host antimicrobial defense, also exert important effects on cell cycle progression and dermal fibroblast proliferation, after interacting with the SRSRY domain of uPAR, confirming our hypothesis that FPRs/uPAR crosstalk plays a crucial role in many pathogenetic aspects of SSc.

### 2.4. FPR/uPAR-Dependent Proliferation of Normal and SSc Primary Dermal Fibroblasts

The data obtained in previous experiments clarified the molecular mechanisms through which FPRs/uPAR crosstalk regulates cell growth and proliferation in a human healthy dermal fibroblast cell line. From then on, we focused our attention on primary dermal fibroblasts from skin biopsies of SSc patients and healthy donors to allow a better understanding of the dynamics of cellular responses to FPR activation and FPRs/uPAR crosstalk in SSc.

First, we compared cell proliferation between normal and SSc fibroblasts to determine whether different growth rates could be found in these cells. To this end, we used fMLF 10^−4^ M, fMLF 10^−8^ M, uPAR_84–95_ peptide 10^−8^ M and WKYMVm peptide 10^−8^ M as proliferative stimuli, and FCS as a generic stimulus. As shown in Figure 4, at 24 h after cell stimulation no effects were recorded; at 48 h, SSc fibroblasts showed a significant increase in proliferation in response to FCS, as compared to normal fibroblasts. At 72 h, SSc fibroblasts proliferated more than normal fibroblasts in response to FCS and WKYMVm peptide, while at 144 h SSc fibroblasts showed a significant increase in proliferation in response to FCS, uPAR_84–95_ and WKYMVm peptide, as compared to normal fibroblasts.

These results demonstrate that FPR activation and FPRs/uPAR interaction are implicated in primary fibroblast proliferation and can confer a higher proliferative phenotype to SSc fibroblasts.

### 2.5. Effect of Selective Inhibitors of FPRs/uPAR Crosstalk on SSc Fibroblast Proliferation

Recently, our group reported new promising lead compounds for pharmaceuticals in cancer and inflammation. These compounds, named C6 and C37, are uPAR inhibitors identified through structure-based virtual screening (SB-VS) of the National Cancer Institute (NCI, National Health Institutes, Bethesda, MD, USA) Diversity Set II [44].

C6 and C37 inhibit uPAR interaction with FPRs by targeting the uPAR chemotactic domain comprising aa 88–92 (SRSRY sequence), which can mediate uPAR’s interaction with FPRs. C6 interacts with S88 and R91, preventing the structural interaction between uPAR and FPRs; C37 interacts only with the R91 side chain, thus slightly inhibiting the structural FPRs/uPAR interaction [45].

Therefore, we aimed to investigate whether the inhibition of the structural and functional interaction between FPRs and uPAR could affect the proliferative activity of fibroblasts from SSc patients. These cells were incubated, at different time points, with DMSO alone (vehicle control) and specific agonists, alone or in the presence of 20 μM of C6 and C37. As shown in Figure 5, C6 dramatically reduced the cell proliferation, whereas C37 did not exert any inhibitory effects on SSc fibroblast proliferation.

Hence, the inhibition of the FPRs/uPAR interaction by C6 showed that FPRs are not able to elicit cell proliferation alone, but they engage uPAR on the cell surface to trigger fibroblast proliferation.

## 3. Discussion

The role of FPRs associated with inflammation and the development of fibrosis has been increasingly investigated in recent years [17,46].

Among elucidated molecular aspects of the pathogenesis of Systemic Sclerosis (SSc), our group has previously suggested that FPRs play an important role in the induction of the myofibroblast phenotype and in the generation of Reactive Oxygen Species (ROS). The dysfunctional behavior of FPRs, in the context of SSc-associated fibrosis, could be due to their structural interaction with a specific cleaved form of uPAR, DII-DIII-uPAR_88–92_, which is overexpressed in SSc skin fibroblasts. Therefore, targeting the FPRs/uPAR crosstalk represents a suitable therapeutic approach to prevent fibrosis progression.

Hence, we examined the FPR’s ability to also elicit cell growth and proliferation in normal dermal fibroblasts, using the BJ cell line as a model.

We demonstrated that FPR stimulation with specific agonists could induce fibroblast proliferation. FPRs recognize many agonists, which comprise three subtypes, pathogen-derived, host-derived, and synthetic molecules. We focused on E. coli-derived fMLF, WKYMVm, a synthetic hexapeptide isolated from a random peptide library, and uPAR_84–95_, a peptide containing the uPAR-derived ^88^SRSRY^92^ sequence able to interact with FPRs, thus mimicking the effects of FPRs engagement by native uPAR on the cell surface. Our study demonstrated that fMLF can induce fibroblast proliferation through FPR1 and FPR2 activation, while uPAR_84–95_ and WKYMVm peptides also bind FPR3.

Moreover, fibroblast proliferation induced by FPR stimulation was totally abrogated by the pretreatment of BJ cells with selumetinib, a specific ERK1/2 inhibitor. The pretreatment with a Rac1 inhibitor, NSC23766, exhibited effects only in response to uPAR_84–95_ and WKYMVm peptides. These findings indicate that FPRs, by interacting with uPAR, can induce fibroblast proliferation through the activation of ERKs and that Rac1 is involved in the integration of FPR-dependent signaling pathways, unlike what was found in FPRs-mediated ROS production [22]. As shown by the inhibition of WKYMVm and uPAR_84–95_-dependent fibroblast proliferation by NSC23766, Rac-1 appears to be exclusively involved in the proliferative signaling downstream of FPR3.

The analysis of the molecular mechanism underlying FPRs/uPAR network revealed that FPRs, activated by uPAR_84–95_ peptide, were able to regulate cell cycle progression through c-Myc serine 62 phosphorylation and Cyclin D1 upregulation.

After establishing the role of FPRs and their crosstalk with uPAR in the proliferation of a healthy fibroblast cell line, we pursued our studies on primary cultures of skin fibroblasts isolated from biopsies of SSc patients and healthy control subjects. Human fibroblasts isolated from affected skin are an ex vivo organ model of fibrosis with a great potential for investigating the mechanisms underlying SSc-related fibrosis. Since fibroblasts are the effector of SSc-related fibrosis, their utility in experimental assays and their contribution to drug development and clinical trials for SSc have been reported [47].

The proliferative potential, both in basal conditions and in response to FPR agonists, was higher in SSc fibroblasts than in normal fibroblasts, suggesting that SSc progressive fibrosis could be linked to an aberrant activation of FPR signaling induced by the upregulation of DII-DIII-uPAR_88–92_. It is conceivable that the overexpression of DII-DIII-uPAR_88–92_ on the cell surface of SSc fibroblasts leads to the constitutive activation of FPRs, thus resulting in pathological tissue fibrosis.

Thus, we analyzed the effects of small molecules targeting FPRs/uPAR crosstalk, C6 and C37 on SSc fibroblast proliferation. In our previous work, C6 and C37 had already been tested on cancer cells and we had observed a significant inhibition of cell proliferation in RAS-mutated cells after C37 treatment, while C6 did not exert any effect [45]. Conversely, SSc fibroblast proliferation was exclusively inhibited by C6, while C37was inactive. These results could be explained by the different binding activities of C6 and C37 to uPAR; C6 completely prevents the structural interaction between uPAR and FPRs, whereas C37 only slightly inhibits the structural FPRs/uPAR interaction. Additionally, C37, with a slightly better affinity, and C6 can inhibit uPAR binding to vitronectin (VN); indeed, C6 mimics VN itself, extending into the uPAR-VN binding site, while C37 entirely fills the VN recognition pocket of the receptor [45]. Thus, the differences in uPAR inhibition by specific small molecules indicate that uPAR contacts different molecular partners on the cell surface in chronic inflammation and tumorigenesis. Based on our results, we could hypothesize that FPRs/uPAR crosstalk is crucial in chronic inflammation, while uPAR’s binding to VN and its crosstalk with EGFR is preferentially used by cancer cells to proliferate.

In our previous work, the treatment of SSc fibroblasts with C37 resulted in a strong inhibition of ROS production, whereas C37 did not exert any effects on cell proliferation in the present data [22]. Hence, C37 inhibits the oxidative stress, while C6 blocks the proliferation of SSc fibroblasts. All together, these results indicate that FPR-dependent cell proliferation requires the expression of DII-DIII- uPAR_88–92_, whereas FPR-mediated ROS requires the expression of full-length uPAR including D1, as the formation of multimolecular complex, including vitronectin and integrins, is necessary. In fact, C37 that is active mainly on the uPAR/VN interaction blocks ROS generation, but not cell proliferation in SSc fibroblasts. An illustrated explanation of our hypotheses is reported in Figure 6.

## 4. Materials and Methods

### 4.1. Peptides and Chemicals

The hexapeptide Trp-Lys-Tyr-Met-Val-D-Met-NH_2_ (WKYMVm) was synthesized and HPLC purified (95%) by Innovagen (Lund, Sweden); the peptide uPAR_84–95_ was synthesized by PRIMM (Milan, Italy) and N-Formyl-l-methionyl-l-leucyl-l-phenylalanine (fMLF) was obtained from Calbiochem (La Jolla, CA, USA). Protein concentration was determined with a modified Bradford assay (Bio-Rad Laboratories, Munchen, Germany). ECL Plus was obtained from GE Healthcare (Buckinghamshire, UK). The protease and phosphatase inhibitors were obtained from Calbiochem. Rabbit anti-FPR1, mouse anti-FPR2, rabbit anti-FPR3, rabbit anti-phospho-c-myc, mouse anti-c-myc, rabbit anti-ciclyn D1 were from Santa Cruz Biotechnology (Santa Cruz, CA, USA); mouse anti-β-actin was obtained from Sigma-Aldrich (St. Louis, MO, USA); secondary anti-mouse and anti-rabbit antibodies coupled to HRP were from Bio-Rad (Munchen, Germany). NSC23766 was from Calbiochem, and selumetinib (AZD6244) was from AstraZeneca (Cambridge, UK); C6 and C37 were from the NCI/DTP Open Chemical Repository (Available from: http://dtp.cancer.gov (accessed on 12 August 2013). They were dissolved in dimethyl sulfoxide (DMSO), stored at −20 °C and added to the cell culture at final concentrations, as indicated in the text.

### 4.2. Tissues and Patient Samples

Three females affected by SSc, observed from January 2020 to December 2023 in the Day Hospital of Department of Translational Medical Sciences of the University of Naples Federico II, were classified according to the American College of Rheumatology Criteria as having limited cutaneous (*n* = 2) or diffuse cutaneous (*n* = 1). The informed consent was signed by the patients enrolled in the study. The mean age of patients was 53 y (range, 48–61 y). Clinically involved skin was defined as values of skin thickness ≥ 2, according to the modified Rodnan skin thickness score. SSc patients admitted for the study were positive for antinuclear antibodies with a speckled pattern evaluated by indirect immunofluorescence. SSc patients with diffuse cutaneous form presented anti-SCL-70 topoisomerase I positivity, and patients with limited cutaneous SSc were positive for anticentromere (CENP-B). SSc patients enrolled in this study had no other overlapping autoimmune, rheumatic and/or connective tissue diseases. The punch biopsy was performed 30 days after all patients were washed out of steroids. Control donors were matched with each SSc patient for age, sex and biopsy site.

The protocol has been approved by the Institutional Review Board (or Ethics Committee) of Naples “Federico II” and is performed according to Good Clinical Practice guidelines and the Declaration of Helsinki.

### 4.3. Cell Cultures

The BJ (human foreskin fibroblasts; ATCC accession number CRL-2522) was from ATCC (LGC Standards, Milan, Italy) and were grown in DMEM (Life Technologies, Carlsbad, CA, USA) with 10% FBS. BJ cells were obtained from ATCC at the 6th passage, subcultured and frozen in stock vials; they were used between the 1st and 10th passage in culture.

Primary skin fibroblasts isolated from punch biopsies of SSc and healthy skin were mechanically dissociated under a light microscope and trypsinizated, as previously described [31,48]. Cells were plated and cultured in monolayer in DMEM (Life Technologies) supplemented with 10% heat inactivated FBS (Life Technologies), 100 U/mL penicillin G sodium and 100 mg/mL streptomycin sulfate, at 37 °C, in a humidified atmosphere of 5% CO_2_. Fibroblasts from both normal subjects and SSc patients were used between the 3rd and 10th passage in culture.

### 4.4. Fibroblast Proliferation Assay

Human skin fibroblasts were serum-starved overnight using DMEM 0.1% BSA, plated at 5 × 10^3^ cells/well in 96-well plates, and incubated with cell culture medium alone or with specific agonists, fMLF (10^−8^ M), uPAR_84–95_ (10^−8^ M) and WKYMVm peptide (10^−8^ M), or with 10% FBS for 1, 24, 48, 72 and 144 h at 37 °C, 5% CO_2_. At the end of the incubation, 20 μL/well CellTiter-96 was added. After incubation at 37 °C for 2 h, the absorbance was determined by an ELISA reader (Bio-Rad) at a wavelength of 490 nm.

### 4.5. Western Blot

Cells were harvested in RIPA lysis buffer (20 mM Tris-HCl pH 7.5, 150 mM NaCl, 1 mM Na_2_ EDTA, 1 mM EGTA, 1% NP-40, 1% sodium deoxycholate, 2.5 mM sodium pyrophosphate, 1 mM β-glycerophosphate, 1 mM Na_3_VO_4_ and 1 µg/mL leupeptin) supplemented with a cocktail of proteases and phosphatases inhibitors. Fifty micrograms of protein were electrophoresed on a 10% SDS-PAGE and transferred onto a polyvinylidene fluoride membrane. The membrane was blocked with 5% nonfat dry milk and probed with specific antibodies: rabbit anti-phospho-c-myc (1 μg/mL), mouse anti-c-myc (1 μg/mL), rabbit anti-ciclyn D1 (1 μg/mL) and mouse anti-actin (0.5 μg/mL). Finally, washed filters were incubated with HRP-conjugated anti-rabbit or anti-mouse Abs. The immunoreactive bands were detected by a chemiluminescence kit and quantified by densitometry (ChemiDoc XRS, Bio-Rad).

### 4.6. Statistical Analysis

All statistical analyses were performed using GraphPad Prism 5.0 software (GraphPad). All the experiments were performed at least in triplicate. The results are expressed as mean ± SEM. Values from groups were compared using a paired Student’s *t*-test [49]. Differences were considered significant when *p* < 0.05.

## 5. Conclusions

The dermal fibroblasts are crucial executors of wound healing, and their proliferation is a key determinant during the cascade of healing. Our results suggested that the FPRs/uPAR axis is involved in dermal fibroblast proliferation and cell cycle progression. The proliferation of dermal fibroblasts is dysregulated in SSc, and FPRs/uPAR targeting may be a suitable strategy for anti-fibrotic intervention. In fact, the inhibition of FPRs/uPAR by specific small molecules blocks dermal fibroblast proliferation. These findings will provide new targets and strategies for clinical interventions in chronic skin wounds and fibroproliferative disease.

## Figures and Tables

**Figure 1 ijms-25-03156-f001:**
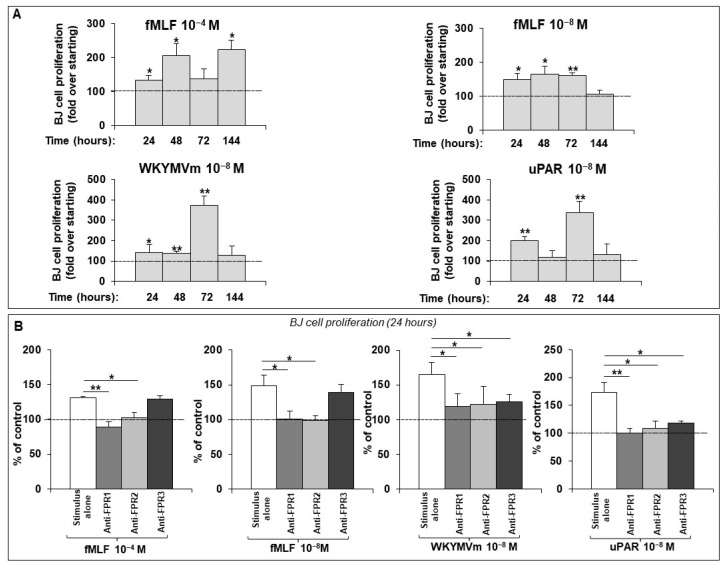
(**A**) Effects of fMLF, WKYMVm and uPAR_84–95_ synthetic peptides on BJ cell proliferation. Values are mean ± SEM of three experiments. * *p* < 0.05, ** *p* < 0.001. (**B**) Effects of anti-FPR1, anti-FPR2 and anti-FPR3 Abs on BJ cell proliferation. BJ cells were treated with medium alone (100%) or medium with stimulus (white columns), in the presence of anti-FPR1 (grey columns), anti-FPR2 (light gray columns) and anti-FPR3 (dark grey columns) Abs. Results are expressed as percent increase in optical density value over untreated cells (dashed lines). Values are mean ± SEM of three experiments. * *p* < 0.05, ** *p* < 0.001.

**Figure 2 ijms-25-03156-f002:**
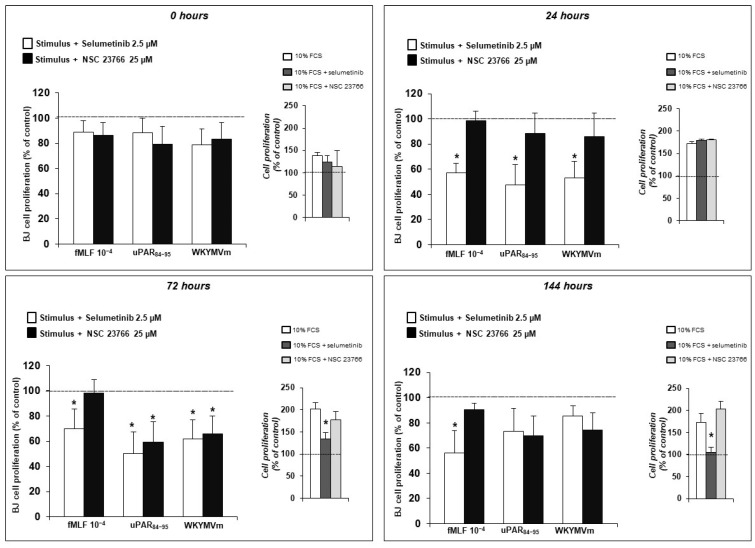
BJ cells were treated with fMLF, WKYMVm peptide, or uPAR_84–95_, in the presence of selumetinib (white columns) and NSC23766 (black columns). Medium alone values (negative control) were subtracted, and results were expressed as percent increase in optical density value over stimulus alone. Values are mean ± SEM of three experiments. * *p* < 0.05. FCS-treated cells were examined in parallel, as controls, and are shown in insets.

**Figure 3 ijms-25-03156-f003:**
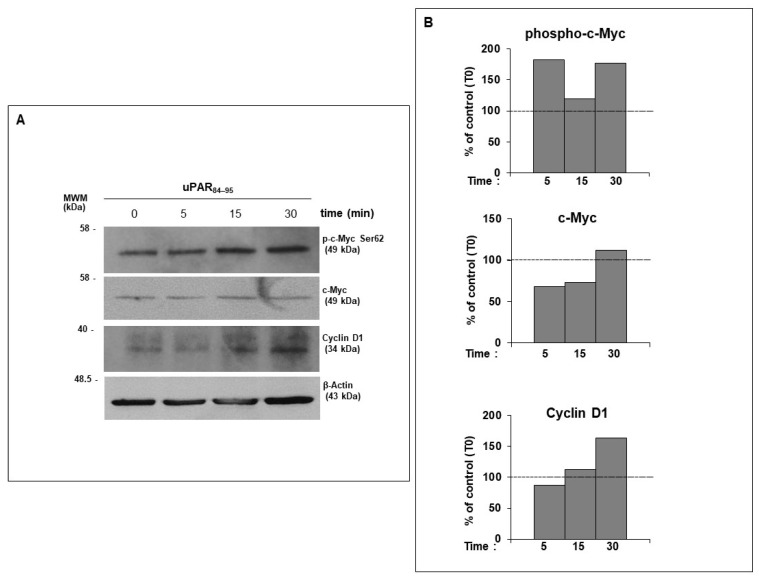
(**A**) BJ cells, treated with uPAR_84–95_ peptide for 0, 5, 15 and 30 min, were lysed and subjected to Western blot analysis with anti-phospho-c-Myc, c-Myc and Cyclin D1 Abs and anti-β-actin A. (**B**) Densitometric analysis of the Western blot bands has been performed using ImageJ’s gel analysis software (version 1.53m; National Institute of Health, Bethesda, MD, USA); after normalization by β-actin, the values were expressed as a fold change over the control (T0).

**Figure 4 ijms-25-03156-f004:**
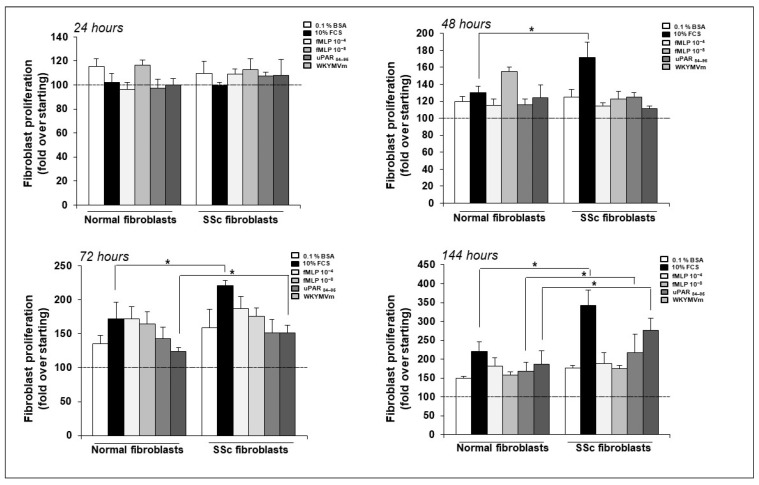
Normal and SSc fibroblasts were treated with medium containing 0.1% BSA (negative control), 10% FCS (positive control), fMLF, WKYMVm peptide and uPAR_84–95_ peptide. Results are expressed as percent increase in optical density value over starting (T0). The statistical significance was determined by comparing SSc to normal fibroblasts. Values are mean ± SEM of three experiments. * *p* < 0.05.

**Figure 5 ijms-25-03156-f005:**
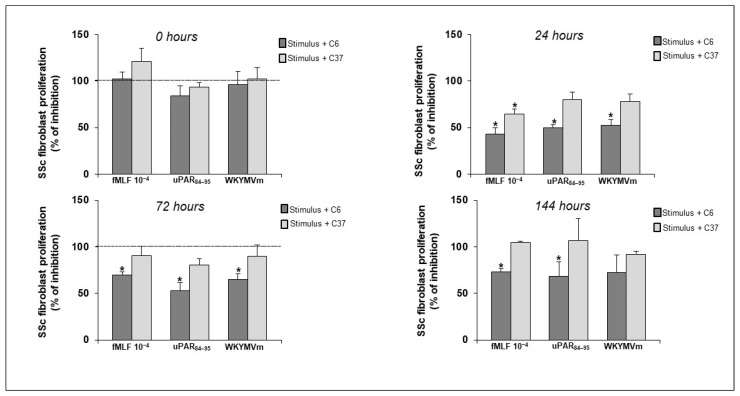
SSc fibroblasts were treated with fMLF, uPAR_84–95_ and WKYMVm peptide, in absence or in presence of C6 (dark grey columns) and C37 (light grey columns) and with DMSO alone, as a vehicle control. After subtraction of OD obtained with DMSO-loaded cells from all obtained OD values, results were expressed as percent decrease in stimulus plus compounds over stimulus alone (100%). Values are mean ± SEM of three experiments. * *p* < 0.05.

**Figure 6 ijms-25-03156-f006:**
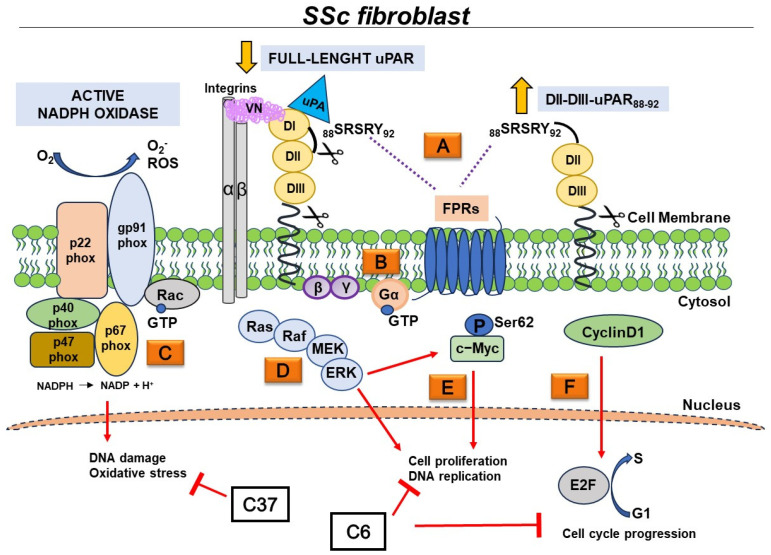
Schematic cartoon illustrating pathogenetic signaling pathways of FPRs in SSc fibroblasts. (**A**) Two forms of uPAR are present on the cell surface: full-length and cleaved uPAR, each specifically interacting with one or more transmembrane partners. In SSc fibroblasts, full-length uPAR is downregulated, whereas DII-DIII- uPAR_88–92_ is overexpressed. FPRs’ crosstalk with both DII-DIII- uPAR_88–92_ and full-length uPAR, upon binding to uPA that induces the exposure of _88_SRSRY_92_ sequence (**B**) The activation of FPRs results in the dissociation of the Gα from the Gβγ subunit. Gα subunit activates GTPases of the Ras superfamily (**C**) FPR activation, via full-length uPAR, is characterized by multimolecular complex that involves vitronectin (VN) and integrins. The downstream signaling event induces NADPH oxidase activation, which contributes to DNA damage and oxidative stress in SSc, through significant ROS generation. C37, a new small molecule able to inhibit FPRs/uPAR crosstalk, inhibits ROS production, after stimulation with specific ligand of FPRs (**D**) uPAR-dependent FPR activation results in the activation of ERK1/2 MAPK pathway (**E**) FPRs/uPAR crosstalk induces phosphorylation of c-Myc at Ser62 residue that is a potent regulator of cell growth, proliferation, and differentiation (**F**) FPRs/uPAR crosstalk induces upregulation of Cyclin D1, which is responsible for G1/S cell cycle progression. The treatment with a new small molecule C6, which completely prevents the structural interaction between uPAR and FPRs, exerts its inhibitory effects mainly on cell growth and proliferation. Downregulated form of uPAR (
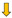
); overexpressed form of uPAR (
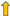
); inhibition (
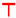
); structural and functional interaction (
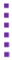
); intracellular signaling cascade elicited by FPRs/uPAR crosstalk (
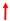
).

## Data Availability

Data is contained within the article.

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
