# Peer review of "The Crosstalk between N-Formyl Peptide Receptors and uPAR in Systemic Sclerosis: Molecular Mechanisms, Pathogenetic Role and Therapeutic Opportunities"

_ijms, 2024, doi:10.3390/ijms25063156_

Round 1

Reviewer 1 Report

Comments and Suggestions for Authors

This study explored the influence of FPR and its external/internal ligands on dermal fibroblast proliferation, along with the effects of C6 and C37 on fibroblast proliferation induced by FPR ligands. The manuscript is well-written, and the methodology is sound.

Major Point:

A notable limitation of this study is the absence of in vivo data, particularly regarding the impact of C6 and C37 on dermal fibroblast proliferation within the context of pathological skin fibrosis, such as in a BLM-induced murine model of Systemic Sclerosis (SSc).

Minor Point:

Another critical aspect to consider is that uPAR-/- mice spontaneously develop SSc-like multiple organ involvement characterized by tissue fibrosis and vasculopathy. This information should be incorporated into the introduction and discussion sections to address the association of FPR with SSc, emphasizing the role of uPAR.

Reviewer 2 Report

Comments and Suggestions for Authors

I think your papers are very interesting and potentially important for understanding the pathogenesis of SSc.

On the other hand, I thought there was a lack of discussion of the involvement of B cells in SSc.

Fukasawa T, Yoshizaki-Ogawa A, Sato S, Yoshizaki A. The role of B cells in systemic sclerosis. J Dermatol. 2024 Feb 6. doi: 10.1111/1346-8138.17134.

Please add a discussion of the relationship with B cells.

If possible, please illustrate the hypothesis derived from the present results.

Round 2

Reviewer 1 Report

Comments and Suggestions for Authors

No further concerns. The authors have made a substantial effort to address the concerns of the reviewer and have significantly improved the paper as a result.